# Effectiveness of Conservative Treatments in Positional Plagiocephaly in Infants: A Systematic Review

**DOI:** 10.3390/children10071184

**Published:** 2023-07-07

**Authors:** Maria Blanco-Diaz, Maria Marcos-Alvarez, Isabel Escobio-Prieto, Marta De la Fuente-Costa, Borja Perez-Dominguez, Elena Pinero-Pinto, Alvaro Manuel Rodriguez-Rodriguez

**Affiliations:** 1Faculty of Medicine and Health Sciences, University of Oviedo, 33006 Oviedo, Spain; blancomaria@uniovi.es (M.B.-D.); mariamarcosalvarez9@gmail.com (M.M.-A.); fuentemarta@uniovi.es (M.D.l.F.-C.); 2Physical Therapy and Translational Research Group (FINTRA-RG), Institute of Health Research of the Principality of Asturias (ISPA), Faculty of Medicine and Health Sciences, University of Oviedo, 33006 Oviedo, Spain; alvaro.manuel.rodriguez@gmail.com; 3Department of Physical Therapy, Faculty of Nursing, Physical Therapy and Podiatry, University of Seville, 41009 Seville, Spain; epinero@us.es; 4Institute of Biomedicine of Seville (IBIS), 41013 Seville, Spain; 5Exercise Intervention for Health Research Group (EXINH-RG), Department of Physical Therapy, University of Valencia, 46010 Valencia, Spain; f.borja.perez@uv.es

**Keywords:** plagiocephaly, physical therapy, positional plagiocephaly, cranial deformity, deformity correction, helmet therapy

## Abstract

Objective: The objective of this study is to analyze conservative treatments implemented to manage positional plagiocephaly in infants. Methods: This is a systematic review conducted according to the Preferred Reporting Items for Systematic Reviews and Meta-Analyses guidelines, performed in the Medline (PubMed), Scopus, Web of Science, and Cochrane databases. Articles were selected according to the eligibility criteria, regarding the effectiveness of conservative treatments in positional plagiocephaly in infants, published in the last 10 years with a score ≥3 in the PEDro Scale. Results: A total of 318 articles were identified and 9 of them were finally selected. Conclusions: Physical therapy treatment is considered as the first line of intervention in plagiocephaly with non-synostotic asymmetries and manual therapy is the method that obtains the best results within this intervention. In cases of moderate or severe plagiocephaly, helmet therapy can be an effective second-line intervention; however, the best way to prevent this condition is through counseling of parents or caregivers, and early treatment is essential for optimal therapeutic outcomes. The review was registered in PROSPERO (CDR42022306466).

## 1. Introduction

The shape of an infant’s head is among the primary factors that prompt medical consultation [1]. Although the etiology and management of these deformities is diverse, some head shape alterations require surgical treatment, such as the premature fusion of cranial sutures (craniosynostosis). Positional plagiocephaly (PP) is not associated with synostotic problems; it is the flattening of one side of the head produced by an external force continuously applied [2]. Such deformity occurs mainly during the first months after birth, and is majorly affected by the head’s positioning [3]. When a flattening of the skull occurs, the natural tendency of the head turns to this side, pulled by gravity [4]. Plagiocephalic infants exhibit flattening on one side of the back of their head, known as unilateral occipital flattening, accompanied by a bulging on the opposite side of the occiput, referred to as contralateral occipital bulging [5]. During growth, malformations can continue to develop, and the degree of remaining malformation appears to be linked to the quantity of synostotic sutures that are affected [6]. Infants with more severe plagiocephaly may also have asymmetric faces [5]. Among the known risk factors in PP, the main factor is keeping the infant in a supine position for too long. Other factors are assisted childbirth, being the firstborn, being male, infant torticollis, and intrauterine constriction [5,7,8,9,10].

Infants diagnosed with PP may exhibit lower levels of activity when compared to children of the same age [11]. Imaging studies suggest that while children with PP may not have differences in brain volume, they often display asymmetries and notable flattening of brain structures such as the cerebellar vermis or corpus callosum. As a result, they may perform lower in cognitive and motor assessments, such as the BSID-III (Bayley Scales of Infant Development, third edition). The BSID-III yields composite scores reflecting infants’ cognitive, language, and motor development [12,13]. 

The incidence of PP, as reported, varies depending on the definitions and assessment methods used. It appears to be age-dependent, with the highest incidence occurring within the first 6 months of life and a tendency to decrease up to 2 years of age [14]. In 1992, the American Academy of Pediatrics launched the “Back to Sleep Campaign”, which encouraged parents to place their infants in a supine position when sleeping to reduce the risk of sudden infant death syndrome (SIDS) [4,15]. This resulted in a reduction of more than 40% in the incidence of SIDS; however, it also led to a substantial increase of around 600% in the incidence of PP [4]. On a global level, previous studies estimated an incidence of 46.6% [16]. The prevalence of PP in Europe is 37.8% in full-term and in infants with no previous pathologies at 8–12 weeks of life [17].

Early recognition in newborns is very important to prevent medical complications and avoid surgical interventions [6,18]. When a good diagnosis is made, infants with PP do not require referral to a surgical specialist [1]. PP can usually be diagnosed through clinical and physical assessments. It can be managed both surgically and non-surgically. Conservative treatments are usually preferred for cases of PP that are not associated with craniosynostosis [4]. Out of the many conservative treatment options, helmet therapy and postural correction work with physical therapy provides good results. It is important to note that conservative treatments should be personalized and tailored to the individual needs of each baby. Evaluation and supervision by healthcare professionals, such as pediatricians and specialized pediatric physiotherapists, are crucial to ensure the success of conservative treatment. Conservative treatments for plagiocephaly include postural adjustments, physical therapy, massage therapy, and the use of head orthoses [19]. Physical therapy for infants incorporates a range of manipulative techniques, such as the Bobath method, craniosacral therapy, postural treatment, and passive exercises, aiming at strengthening their neck and upper body muscles [20]. Additionally, a maneuver to mobilize the neuromeningeal tissue at the lumbosacral level can be used as a complementary treatment. This technique involves applying manual pressure to shape the base of the skull in the opposite direction of the PP torsion at the skull base. Other techniques, such as those to balance the intracranial membranous tension and a molding technique for decompressing the coronal suture, may also be used [21]. 

Conservative treatments, including repositioning techniques, physical therapy, and the use of orthotic devices, have emerged as viable options to mitigate the severity and progression of PP [4,19]. Understanding the role and effectiveness of these conservative treatments is essential to guide healthcare professionals in making informed decisions regarding intervention strategies. However, despite their widespread utilization, there remains a need to comprehensively evaluate the existing evidence base to determine the efficacy, safety, and long-term outcomes associated with conservative treatments for positional plagiocephaly.

According to the scientific literature, different conservative treatments exist. Therefore, the primary objective of this review is to assess the effectiveness of conservative treatments in positional plagiocephaly in infants.

## 2. Methods

### 2.1. Design

Results are reported according to the Preferred Reporting Items for Systematic Reviews and Meta-Analyses (PRISMA) guidelines [22,23]. The protocol was registered in an international registry for systematic reviews (PROSPERO): CRD42022306466.

A systematic electronic search was performed between September 2022 and January 2023 in the following databases: Medline (PubMed), Scopus, Web of Science, and Cochrane. The aim was to identify studies reporting outcomes on positional plagiocephaly and physical therapy. The summary for the search strategy can be found in Table 1. 

### 2.2. Eligibility Criteria

The authors agreed to conduct the search strategy using the PICOS approach (Population, Intervention, Control, Outcomes, and Study design) [24]. Search strategies included DeCS and MeSH terms. The inclusion criteria employed in this systematic review were designed to ensure the selection of relevant studies that addressed the research question while maintaining a balance between comprehensiveness and feasibility. Therefore, the review focused on (1) experimental studies involving patients diagnosed with positional plagiocephaly, that (2) included in their intervention a conservative treatment option. Additionally, (3) only studies published in the English language were included to facilitate accurate data extraction and analysis, and (4) studies published within the last 10 years (2013–2023) were included to provide an up-to-date assessment of the literature and consider recent advancements. To avoid duplication, duplicate papers and multiple reports from the same study with the same outcomes were excluded from the final selection. By adhering to these predefined criteria and employing rigorous search strategies, we aimed to ensure the inclusion of relevant and high-quality studies in our review.

This approach has enabled the establishment of systematic reviews, randomized controlled trials, and meta-analyses. It also facilitated critical reasoning on different issues [24], and the formulation of the following question of what was the evidence for conservative treatment of plagiocephaly in infants.

### 2.3. Data Extraction

Two authors independently (MM and MB) screened titles and abstracts of all identified records. Differences between researchers were resolved through discussion and mediation by a third researcher (IEP). A standard format was used to assess for inclusion. The first author, publication year, article type, design of the included studies, number of included studies or any disagreements either at this stage or further on in the process were settled by mediation through a third author, as well as study population, interventions under study, outcome measures, main results, and the authors’ conclusions.

### 2.4. Outcomes

Anthropometric assessments and clinical results were registered, including the Cranial Index (CI) [21,25], Cranial Vault Asymmetry Index (CVAI) [19,20,25,26,27,28], the Index of Oblique Diameter Difference (ODDI) [28], and the Posterior Cranial Asymmetry Index (PCAI) [26]. Ratings of deformation were made on the following scale: 0 = none, 1 = mild, 2 = moderate, and 3 = severe by using a scale based on a measure described by Branch et al. [29], the Alberta Infant Motor Scale (AIMS) for the assessment of gross motor development [30], Maximal Cranial Circumference (MCC), Ear Deviation Index (EDI), and Visual Analogue Scale (VAS) in order to assess the perception of the change in head shape, and the Infant Toddler Quality of Life Questionnaire.

### 2.5. Methodological Quality Assessment

The methodological quality of the included randomized controlled trial studies was evaluated using PEDro scale. The PEDro scale (ranged from 0–10) is based on the Delphi list developed by Verhagen et al. [31,32]. Studies scoring 9–10 points on the PEDro scale are considered to have Excellent methodological quality. Studies with a score between 6 and 8 have a Good methodological quality, between 4 and 5 have a Fair methodological quality, and studies scoring below 4 points are considered to have a Poor methodological quality. One author extracted data and a second author checked it. Any disagreements were resolved by consensus and mediation from a third author.

### 2.6. Risk of Bias Assessment

Two independent reviewers assessed each article for potential sources of bias. Each item was rated as having “high risk”, “low risk”, or “unclear risk” of bias. To further validate the results, a sensitivity analysis was conducted to examine the impact of including or excluding studies with a high risk of bias on the primary outcomes [33].

## 3. Results

### 3.1. Study Selection

The initial search in the databases yielded a total of 318 articles from Medline (PubMed), SCOPUS, Web of Science, and the Cochrane Library.

An initial screening produced 168 articles after duplicates were removed (*n* = 149). Subsequently, titles and abstracts were screened for eligibility by two independent authors. A total of 137 studies were removed for not focusing on the topic (*n* = 64), not providing relevant information (*n* = 53), being case reports (*n* = 4), commentaries (*n* = 2), or not meeting the eligibility criteria (*n* = 14). The remaining 31 studies were screened for full-text review by two independent authors, who registered reasons for exclusion. After this process, 22 studies were excluded due to their population. Finally, nine articles were assigned to two different examiners who assessed them independently [20,21,24,25,26,27,34,35,36]. The different phases of the review are illustrated in Figure 1.

### 3.2. Quality of Included Studies

The results for the assessment of the quality of the included studies are presented in Table 2. One studies received a score of two, one study scored six points, one study scored seven points, and, finally, two studies scored nine points.

### 3.3. Risk of Bias Assessment

Results for the risk of bias assessment are shown in Figure 2. Seven important aspects that could affect the bias of the study, including randomization, treatment allocation concealment, blinding of participants and researchers, outcome assessment blinding, completeness of outcome data, selective reporting, and other sources of bias, were assessed. The score was marked as “Low Risk” (represented by “+”), “High Risk” (represented by “−”), or “Unclear Risk” (represented by “?”) for each criterion. In case of any discrepancy, a third reviewer was consulted for resolution.

### 3.4. Main Findings

The characteristics of the included studies are shown in Table 3. A total of 5051 patients with PP were included in this review, varying sample sizes in range from *n* = 24 [28] up to *n* = 4378 [27] participants.

Cabrera-Martos I et al. [35] based their intervention on a conservative approach including positional changes and the use of an orthotic helmet. Pastor-Pons I et al. [21] implemented manual therapy and an educational program for caregivers. Van Wijk et al. [36] compared a helmet therapy with the natural progression of cranial asymmetry without intervention. Kuntz et al. [26] divided participants into two groups, one receiving helmet therapy and the other group without it. Seruya et al. [34] treated patients with helmet therapy. Gonzalez Santos et al. [20] implemented helmet therapy or physical therapy. Di Chiara et al. [28] assessed the change in anthropometric measures pre- and post-pediatric physical therapy. Finally, Steinberg J et al. [27] implemented either conservative management or helmet therapy.

#### 3.4.1. Manual Therapy Techniques

Pastor-Pons et al. [21] implemented an upper cervical spine protocol mobilizing cranial structures to restore ROM. The AROM right rotation improvement was significantly larger in the intervention group than in the control group, 13.4 ± 9.1° and −1.6 ± 9.5° (*p* = 0.000). At baseline, the right AROM was significantly lower in the intervention group. The total cervical rotation AROM increased in both groups. The intervention group improved more than the control group, 29.7 ± 18.4° and 6.1 ± 17.7° (*p* = 0.001), respectively [21]. 

Cabrera-Martos I et al. [30] focused on reducing the biomechanical overload by functionally improving the movement of the joints, mainly the spheno-occipital, the atlanto-occipital synchondrosis, and the sacrum. Over time, a progressive improvement in deformity level was noted in all the treated infants, indicating a lesser degree of observable deformity as assessed with the Argenta scale. The asymmetry at the end of the treatment was minimal, with a score of 0 or 1. Duration of treatment was also significantly shorter (*p* < 0.001) in the intervention group (109.84 ± 14.45 days) compared to the control group (148.65 ± 11.53 days). Additionally, motor behavior was within the normal range (scores above the 16th percentile of the AIMS) in every participant.

The results of the study by Chiara et al. [28] showed improvements in the change in four of the anthropometric measures, performed pre- and post-physical therapy program, being greater in younger children and in the most severe presentations (*p* < 0.05 or *p* < 0.01).

Pastor-Pons et al. [21] used a 10-session program consisting of manual therapy and a caregiver education program aimed at reshaping cranial deformation. CVAI presented a greater decrease in the intervention group (3.72 ± 1.40%) compared with the results of the control group, 0.34 ± 1.72% (*p* = 0.000). CI did not present significant differences between groups. Furthermore, a significant increase in cranial length was found in the intervention group (7.57 ± 2.33 cm) in contrast with the control group (4.25 ± 2.47 cm) (*p* = 0.001). 

Pastor-Pons et al. [25] also showed that, after intervention, the pediatric integrative manual therapy group presented a significant increase in rotation (29.68 ± 18.41°) than the control group (caregivers receiving an evidence-based educational physical therapy program) (6.13 ± 17.69°) (*p* = 0.001). No statistically significant differences were found, although both groups improved neuromotor development.

#### 3.4.2. Helmet Therapy

Van Wijk et al. [36] compared natural evolution in cranial deformation with helmet therapy. For plagiocephaly and brachycephaly, the change score was equal between both groups, with a mean difference of −0.2 (95% confidence interval −1.6 to 1.2, *p* = 0.80) and 0.2 (−1.7 to 2.2, *p* = 0.81), respectively. A total of 10 out of 39 (26%) participants in the helmet therapy group achieved full recovery, as well as 9 out of 40 (23%) participants in the natural evolution group (odds ratio 1.2, 95% confidence interval 0.4 to 3.3, *p* = 0.74).

Kunz F et al. [26] showed that the largest reduction in head asymmetry was observed in the intervention group when comparing the changes in the symmetry-related variables in all of the three groups (∆T1−T3).

Seruya M et al. [34] divided patients into seven groups based on their age at the start of treatment. Significant differences were found between groups when assessing the final transcranial difference (*p* < 0.05). The transcranial difference median rate of change ranged from 0.41 to 0.93 mm/week. Age at treatment baseline was negatively related to the rate of change in transcranial difference (r = −0.88, *p* < 0.05). At the end of the treatment, cranial symmetry had improved in all groups.

Steinberg J et al. [27] achieved a complete correction in 92.8% of participants. A total of 77.1% of the conservatively managed participants achieved complete correction with repositioning therapy. A subset of participants were transitioned with helmets (crossover group) because they failed to improve. The remaining 7.1% ultimately failed to achieve complete correction with continued conservative therapy. Complete correction was achieved in 95.0% of these 1531 total participants who underwent helmet therapy. There were no differences in outcomes between crossover patients who transitioned to helmet therapy after a mean of 4.1 ± 1.4 months of conservative therapy and those who received helmet therapy as first line treatment (96.1% versus 94.4%; *p* = 0.375). 

The results in Gonzalez-Santos et al. [20] indicated that the initial CVAI for the entire sample was 10.69% (SD = 5.58), with a CVAI of 9.62% (SD = 5.59) for the group with helmet therapy and 11.59% (SD = 5.51) for the group with physical therapy treatment (*p* = 0.228). Upon final evaluation, the CVAI dropped to 4.07% (SD = 2.26) in the cranial helmet group and 5.85% (SD = 3.60) in the physical therapy group. No significant statistical differences were found between groups (*p* = 0.70).

**Table 3 children-10-01184-t003:** Characteristics of the included studies.

Author	Study Types	Population	Objective	Intervention	Outcome Measures	Results and Author Conclusions	Side Effects
Cabrera-Martos I et al. 2016 [35]	RCT	46 infants with severe nonsynostotic plagiocephaly referred (types 4–5 of Argenta scale).	Assess the effects of a therapeutic approach based on manual therapy as an adjuvant option on treatment duration and motor development in infants with severe nonsynostotic plagiocephaly.	IG: manual therapy protocol added to standard treatment. CG: standard treatment (conservative approach including changing positions and helmet therapy).	Treatment duration and motor development assessed with the AIMS at baseline and at discharge.	End-treatment asymmetry was minimal, with a score of 0 or 1 in both groups. Manual therapy combined with usual treatment significantly reduced treatment duration.	No adverse effects were observed during the intervention.
Van Wijk RM et al. 2014 [36]	RCT	84 infants aged 5 to 6 months with moderate to severe skull deformation, who were born after 36 weeks of gestation and had no muscular torticollis, craniosynostosis, or dysmorphic features.	To determine the effectiveness of helmet therapy for skull deformation compared with the natural course of the condition in infants aged 5–6 months.	IG: Six months of helmet therapy. CG: natural course of skull deformation.	Change scores for plagiocephaly (oblique diameter difference index) and brachycephaly (cranio proportional index) Infant Toddler Quality of Life Questionnaire.	Equal effectiveness of helmet therapy and skull deformation following its natural course.	All parents reported one or more side effects. Problems with acceptance of the helmet, skin irritation, augmented sweating, unpleasant odor of the helmet, pain associated with the helmet, and feeling hindered from cuddling their child.
Pastor-Pons I et al. 2021 [21]	RCT	34 neurologically healthy subjects aged less than 28 weeks old with a difference of at least 5 mm between cranial diagonal diameters	To assess how effective it is to incorporate manual therapy techniques specific for pediatrics, to a caregiver education program in anthropometric cranial measurements and the subjective parental perception of the cranial shape change in infants with PP.	IG: manual therapy plus a caregiver education program. CG: education program exclusively.	Cranial shape was evaluated using CI and CVAI. Parental perception of change was assessed using a visual analogue scale.	CVAI presented a greater decrease in IG group compared with the CG. CI did not present significant differences between groups. Manual therapy led to a more positive parental perception of cranial changes	No adverse effects were reported.
Pastor-Pons I et al. 2021 [25]	RCT	34 neurologically healthy subjects aged less than 28 weeks old with a difference of at least 5 mm between cranial diagonal diameters	To analyze the effect of manual therapy on the active cervical rotation and in the neuromotor development in a sample of children with PP.	IG: educational approach and specific protocol based on pediatric integrative manual therapy for 10 weeks. CG: educational approach, therapeutic exercise to reduce preference of position and for motor development.	CI, CVAI, neuromotor development: evaluated using AIMS, Cervical AROM to each side (registered by a photographic image from above).	Incorporating manual therapy into a caregiver education program is linked to an improved outcome regarding neck mobility in PP. No outcome differences in neuromotor development were shown	No adverse effects were reported.
Steinberg J et al. 2015 [27]	CT	4378 patients assessed for deformational plagiocephaly and/or deformational brachycephaly were assigned to conservative (repositioning therapy, *n* = 383; repositioning therapy plus physical therapy, *n* = 2998), or helmet therapy (*n* = 997).	To assess the effectiveness of conservative (repositioning therapy with or without physical therapy) and helmet therapy, and identify factors associated with treatment failure.	IG: Helmet therapy. CG: Conservative therapy: repositioning therapy, repositioning therapy plus physical therapy.	Cranial vault anthropometrics were obtained using the STAR scanner Laser Data Acquisition System. Cranial ratio and diagonal difference measurements using CVAI	Conservative treatment and helmet therapy were found to be effective for correcting positional cranial deformation in 92.8 percent of infants. A total of 77.1% of the conservatively treated patients achieved complete correction. The 15.8% of the initial cohort were transitioned to helmets because they failed to improve. The remaining 7.1% ultimately failed to achieve complete correction with continued conservative therapy. Delaying the initiation of helmet therapy for a trial of conservative treatment does not preclude complete correction, provided that the helmet therapy is begun while brain growth is ongoing, and patients are compliant.	No adverse effects were reported.
Di Chiara A et al. 2019 [28]	CT	24 patients diagnosed of non-synostotic asymmetry, a minimum of 1 and a maximum of 18 months.	The authors assessed the modification of anthropometric measurements before and after a pediatric physical therapy program in a sample of patients with non-synostotic skull asymmetry.	16 sessions (40 min) once a week, 4 months of physical therapy: combination of exercises and manipulative procedures to reduce positional preference, musculoskeletal disorders, and cranial deformity.	Argenta scale, Oblique Diameter Difference Index (ODDI), Cranial Proportional Index (CPI), or Cephalic Ratio (CR), CVAI.	A pediatric physical therapy program should be considered as first line of intervention for any non-synostotic asymmetry, independently from their initial severity or their age of first referral.	No adverse effects were reported.
Kunz F. et al. 2019 [26]	RCT	45 infants with DP (CVAI) > 3.5%	Evaluate the long-term outcomes of children who underwent helmet therapy because of DP in the first year of life.	IG: helmet therapy (32 infants with DP). Untreated group: 13 infants with DP who did not undergo helmet therapy. CG: 18 infants without visible head asymmetries and a CVAI ≤ 3.5%.	3D-stereophotogrammetry, CVAI, EO, PCAI.	Head orthosis therapy in patients with DP leads to significantly better long-term outcomes compared with active repositioning or physiotherapy alone	No adverse effects were reported
Seruya M. et al. 2013 [34]	CT	346 infants with deformational posterior plagiocephaly underwent helmet therapy. Patients were stratified by age.	To assess the relationship between age at initiation of helmet therapy for deformational plagiocephaly and the rate of correction.	IG: Helmet therapy use between 20 and 23 h/day. Helmets were adjusted every 2 weeks by removing or adding material to the foam liner.	Transverse cranial diameters’ difference. Average number of hours of helmet use per day.	The correction rate of plagiocephaly with helmet therapy decreases with increasing infant age; after 32 weeks, there is a slow and relatively constant rate of change. Improvement can still be achieved in infants older than 12 months.	No adverse effects were reported
Gonzalez Santos J. et al. 2020 [20]	CT	60 infants	To assess the effect CHT and PT on the evolution and cranial asymmetry in a group of infants with cranial deformities.	IG: Physical Therapy treatment (26 babies). CG: helmets (22 patients)	CVA, CVAI, Early Childhood Psychomotor Development Scale, Posture developmental quotient.	The results indicated that both therapies (CHT and PT) led to improvements in cranial deformity, with no statistically significant differences between both treatments. Despite these findings, the authors suggest using a combination of both techniques, starting with PT, and supplementing it with CHT for patients with a higher CVAI.	No adverse effects were reported

AIMS: Alberta Infant Motor Scale; AROM: Active Rotational Range of Motion; CG: Control Group; CHT: Cranial Helmet Therapy; CI: Cranial Index; CT: Clinical Trial; CVAI: Cranial Vault Asymmetry Index; DP: Deformational Plagiocephaly; IG: Intervention Group; PCAI: Posterior Cranial Asymmetry Index; PIMT: Pediatric Integrative Manual Therapy; PP: Positional Plagiocephaly; PT: Physical Therapy; RCT: Randomized Controlled Trial.

## 4. Discussion

This systematic review gives an overview of all previously published studies for conservative treatments in PP, and highlights the paucity of studies in the literature on the optimal techniques for treating PP. The degree of correction in PP is influenced by age and the type of treatment [37]. Being aware of the recommendations to diagnose and manage PP is determinant in healthcare providers [38]. The program for educating caregivers comprised a series of recommendations based on the literature and included suggestions for motor, sensory, and repositioning stimulation of non-preferred sides and prone positions. Parents received guidance from a physical therapist specialized in pediatrics and were given an informative booklet containing basic guidelines [20,39,40,41]. The related literature also emphasizes the significance of early screening and provides prevention techniques such as head repositioning and positioning the infant in alternating right/left supine positions, especially during the initial 2–8 weeks of life when the skull is most vulnerable to external forces [40]. It is generally accepted that conservative therapy, including repositioning and physical therapy, is appropriate as an initial treatment, and that cranial orthotics should only be considered if no improvement is seen [39,42,43].

The Congress of Neurological Surgeons Systematic Review indicated that physical therapy is a more effective approach than educational strategies for repositioning in PP [43]. The American Association of Pediatrics discourages the use of positioning pillows in an infant’s sleep environment; therefore, the Plagiocephaly Guidelines Committee recommends the use of physical therapy over positioning devices [44].

In conservative management strategies, how severe the deformity is measured by cranial ratio and diagonal difference, torticollis that persists over 6 months, and a delay in neuromuscular development were risk factors, along with age and the level of compliance [22]. Physical therapy should be considered as the first line of intervention for any non-synostotic asymmetries, no matter their severity or first referral age [28,39]. Manipulative treatment is beneficial as it acknowledges the significance of managing the body as a single functional unit, promoting homeostatic processes [36]; the sooner it is applied, the more effective it is. Early strategies both increase the efficacy of the treatment and reduce the worsening rate [28]. Adding manual therapy to the usual management plan leads to a shorter treatment period for infants with a severe plagiocephaly, considered as non-synostotic [36]. Treatments with functional manual therapy manage to improve the asymmetries presented by children younger than 6.5 months old with PP [45].

Helmet therapy is critically questioned with regard to cost-effectiveness and possible commercial involvements [46]. Benefits for children with severe PP are widely accepted [47,48,49]. It is considered, but not validated, that an infant’s cranial deformity cannot be corrected with helmet therapy after 1 year. The literature on the efficacy of helmet therapy for PP suffers from limited study power, subjective outcome measures, and variations in the duration of helmet use and patient compliance. Therefore, the question of how the correction rate is affected by the age of helmet treatment initiation and treatment efficacy in older children remains unclear [35]. Guidelines suggest the use of cranial orthotic helmets for cases of moderate to severe plagiocephaly that present at a later stage of age and for infants with persistent moderate to severe plagiocephaly after a course of conservative treatment (repositioning and/or physical therapy). Tamber et al. [41] indicated that there is a considerable body of non-randomized evidence that shows more significant and faster improvement of cranial asymmetry in infants with PP treated with a helmet than with conservative therapy, especially if the asymmetry is severe, and indicated that helmet therapy is applied during the appropriate period of infancy. However, van Cruchten et al. [44] argued that a combination of physical therapy and helmet therapy provides long-term beneficial results. Generally, infants with more severe deformities and those who begin using helmets early in infancy tend to achieve better correction, and in some cases, even normalization of head shape [39]. To estimate the link between the rate of correction and age at initiation of helmet therapy for deformational plagiocephaly, Seruya et al. [34] showed that with increasing infant age, the success rate of plagiocephaly correction with helmet therapy decreases after 32 weeks, and the rate of improvement slows down and becomes relatively stable. However, improvement can still be attained in infants who are older than 12 months of age [34]. Several studies claim to demonstrate that head orthosis therapy in patients with PP leads to significantly better long-term outcomes compared to active repositioning or only physical therapy [26,50,51,52]. Even after the completion of head orthosis therapy, residual cranial asymmetries continue to improve over time. The results match with those of Kim MJ et al. [53], where patients with PP had a higher risk of developing lateral crossbites, with the likelihood being greater on the side opposite to the posterior flattening. In patients with PP who do not receive head orthosis therapy, facial asymmetries are more commonly observed. The results of the study of Gonzalez Santos et al. [20] showed that in both treatment groups, the CHT group and PT group, the infants showed a progressive improvement in their performance indexes over the 5-month period between the first and the last assessments. Additionally, results showed no statistically significant variations between the different treatment groups. Kunz et al. [26] discussed that the use of a head orthosis is a suitable option for infants with PP when it comes to reducing cranial asymmetry and diminishing developing facial asymmetries. Despite that, based on the similar outcomes between the natural progression of skull deformation and helmet therapy, combined with the high incidence of adverse effects and the high cost of the latter, Van Wijk, B et al. [36] favored against the use of helmets as a standard treatment for moderate to severe cranial asymmetry in healthy infants. Some authors consider the use of cranial orthotic molding helmets to be appropriate because they achieve complete correction in 95.0% of patients, with no difference in outcome between patients who received helmet therapy after failure of conservative therapy compared to those who received helmets as initial treatment [27,54].

As for the aesthetic deficits, the literature shows that in non-operated patients these can be significant [55]; however, even in infants treated at a very young age, many authors state the possible risk of malformation. Moreover, severe residual deformities are directly related to the age at which the infant is managed [56]. Early surgical treatment does not definitively ensure aesthetic and functional results. Primary surgery should be postponed in infants affected by anterior PP without signs of cranial hypertension, until their bone growth is completed, between 5 and 7 years old, so that it will be possible to achieve an optimal aesthetic result after a single surgical intervention [6]. 

Studies suggest that the cognitive abilities of children treated for PP with or without physical therapy or helmet therapy are not altered in their growth (2–7 years old) [57]. 

Current treatments for infants with PP are usually effective in children without complications [58], although not all infants get better [59]. It is difficult to determine the gold standard therapy to achieve better results due to the lack of standardized measurement systems, and the scarce quality of the scientific literature. However, parental counseling, correct decubitus position, and manual treatment are considered useful and low-cost interventions for families [60].

Previous reviews [10,14,16,60] have addressed causes, prevalence, risk factors, diagnosis, and the management of plagiocephaly. De Bock et al. [10] showed the existence of conflicting evidence that makes it challenging to identify potential risk factors, and Bialocerkowski et al. [14] presented the high heterogeneity found regarding prevalence rates in the existing literature, outlining the presence of conflicting evidence that is proof of the need to develop systematic reviews such as the one conducted in this study. Furthermore, Ellwood et al. [60] presented results regarding the management of PP that are in line with the ones of this review, showing considerable evidence to use manual therapy in favor of helmet therapy. 

This study is not exempt from limitations. A potential publication bias could have been incurred in this study due to the limited number of resources accessed during the systematic review process. Additionally, language bias may have been introduced in this study due to the inclusion of only studies published in the English language. The decision to include studies solely in English was made based on resource constraints and the assumption that English language publications are more widely accessible and representative of the available evidence. 

Future reviews could consider including studies in multiple languages to minimize language bias and enhance the generalizability of findings. The current review underscores the need for further research in this field. More studies comparing different treatment modalities, evaluating long-term outcomes, and assessing cost-effectiveness are warranted. Critically evaluating the existing evidence, addressing limitations, and identifying future research directions will contribute to the ongoing development of evidence-based management strategies for infants with PP.

## 5. Conclusions

For any non-synostotic asymmetry, a pediatric physical therapy program should be considered as the first-line intervention. Among physical therapy methods, manual therapy has been shown to produce the best results, particularly when combined with counseling for parents or caregivers, which can lead to even greater benefits. Repositioning therapy is the main preventive measure against cranial deformities. It is suggested that for infants who have moderate to severe plagiocephaly that presents itself at a later stage, or for those who still have persistent moderate to severe plagiocephaly even after undergoing conservative treatments, helmet therapy is a recommended solution. 

Orthotic treatment for children can be initiated after six months of age, but starting treatment at a later age may result in a lower therapeutic success. Surgical intervention may be necessary if there are aesthetic or functional issues that do not improve with other treatments. The age of initiation of treatment should be early, as this will result in greater efficacy and a lower rate of worsening. More research is needed on physiotherapy treatment and its results.

## Figures and Tables

**Figure 1 children-10-01184-f001:**
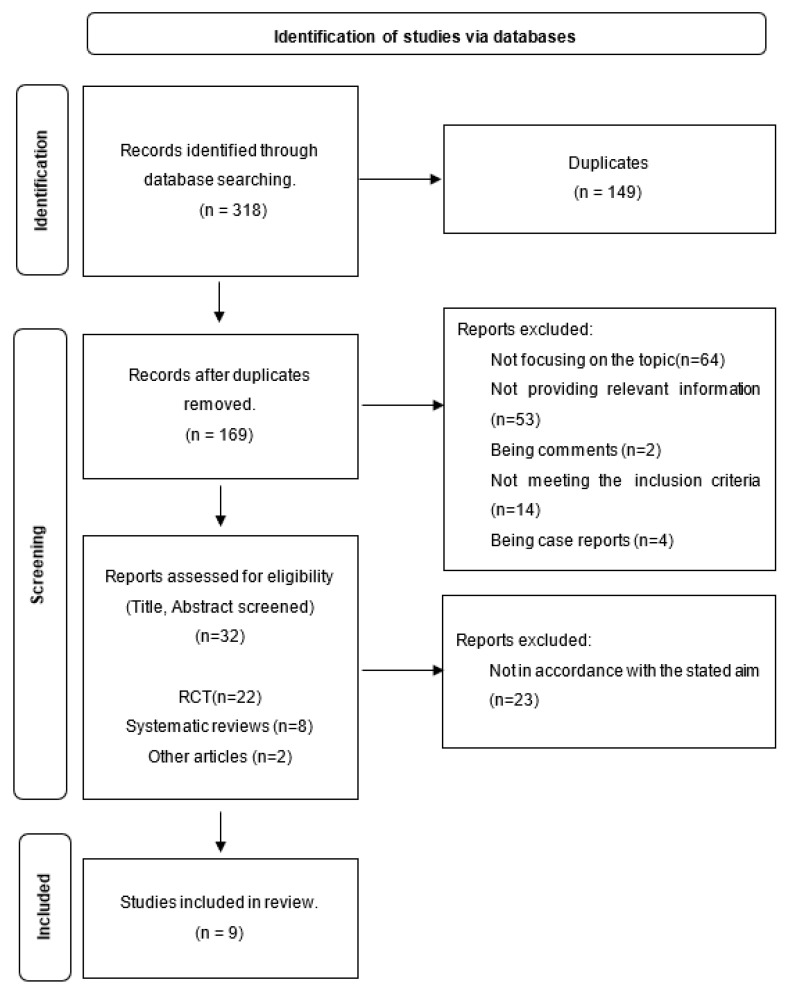
PRISMA Flow Diagram.

**Figure 2 children-10-01184-f002:**
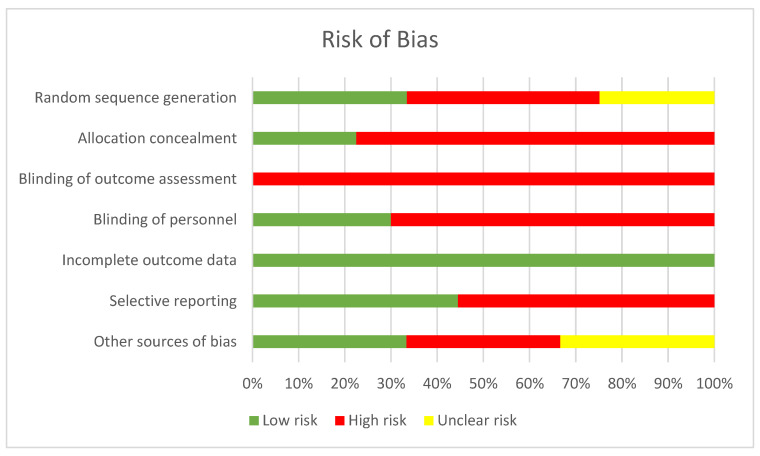
Risk of bias graph of studies.

**Table 1 children-10-01184-t001:** Search strategy used in databases.

PubMed	I. ((Plagiocephaly[title/abstract]) OR (“Plagiocephaly, nonsynostotic” [Mesh])) AND (Physical Therapy). II. (Plagiocephaly [title/abstract]) III. ((“Plagiocephaly/Diagnosis”[Mesh] OR “Plagiocephaly/pathology” [Mesh] OR “Plagiocephaly/Rehabilitation” [Mesh] OR “Plagiocephaly/Surgery”[Mesh] OR “Plagiocephaly/Therapy”[Mesh])) AND (Physical Therapy [Title/Abstract]) IV. ((Argenta[Title/Abstract]) AND (Plagiocephaly[Title/Abstract])
SCOPUS	(ABS (Plagiocephaly) AND ABS (Physical Therapy) OR ABS (“Physical Therapy) OR ABS (“Physical Therapy”))
Web of Science	Plagiocephaly(Topic) AND “Physical Therapy” OR Physical Therapy (Topic)
Cochrane Library	I. Plagiocephaly Title Abstract keyword AND Physical Therapy OR Physical Therapy Title Abstract Keyword

**Table 2 children-10-01184-t002:** Results for the methodological quality assessment of the included randomized controlled trial studies.

Study	Criterion											
	1 *	2	3	4	5	6	7	8	9	10	11	Total
Cabrera-Martos I et al. [35]	x	x	x	x	-	-	x	x	x	x	x	8/10
Van Wijk R M et al. [36]	x	x	x	x	-	-	x	x	x	x	x	8/10
Pastor-Pons I et al. [21]	x	x	-	x	-	-	x	x	x	x	x	7/10
Pastor-Pons I et al. [25]	x	x	-	x	-	-	-	x	x	x	x	6/10
Kunz F et al. [26]	x	-	-	-	-	-	-	x	-	-	x	2/10

Criterion in the PEDro scale: 1 = eligibility criteria; 2 = random allocation of subjects; 3 = allocation concealed: 4 = baseline comparability of important measures; 5 = blinding of subjects; 6 = blinding of therapists; 7 = blinding of assessors; 8 = measures obtained for >85% subjects; 9 = intention to treat analysis; 10 = between-group statistical comparisons; 11 = point measures and measures of variability. * Does not contribute to the total PEDro score. A score of ‘x′ indicates that the criterion is met while a score of ‘-’ indicates that the criterion is not met.

## Data Availability

Not applicable.

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
