# Peer review of "Effectiveness of Conservative Treatments in Positional Plagiocephaly in Infants: A Systematic Review"

_children, 2023, doi:10.3390/children10071184_

Round 1
Reviewer 1 Report
Dear Author
This study is a valuable review investigating effective treatment modalities for Positional Plagiocephaly in infants. On the other hand, there are a number of points that need to be revised, see below.
Major points
l The description of the results is too long. It is recommended to summarize the results obtained for each item.
l 'Physiotherapy' and 'physical therapy' are used in this study. Please unify with one of them.
l Abstract L25-27: In cases of moderate or severe plagiocephaly in elderly patients, helmet therapy can be an effective second-line intervention, however, the best way to prevent this condition is through counseling of parents or caregivers, and early treatment is essential for optimal therapeutic outcomes.
→The intention of 'elderly' being mentioned here cannot be read from the text. Please remove it from the abstract or provide further details in the text.
Author Response
This study is a valuable review investigating effective treatment modalities for Positional Plagiocephaly in infants. On the other hand, there are a number of points that need to be revised, see below.
RESPONSE: Authors would like to thank the Reviewer for the feedback and suggestions. We appreciate the consideration and we hope we can meet the requirements raised by the Reviewer.
Major points
l The description of the results is too long. It is recommended to summarize the results obtained for each item.
RESPONSE: Authors would like to thank the Reviewer for the valuable feedback. We agree that results should be summarized for each item, so we have considerably reduced the content found in our Results section, in lines 161 to 309.
l 'Physiotherapy' and 'physical therapy' are used in this study. Please unify with one of them.
RESPONSE: Authors would like to thank the Reviewer for the suggestion. We have unified terms throughout our manuscript and opted to choose “Physical Therapy”.
l Abstract L25-27: In cases of moderate or severe plagiocephaly in elderly patients, helmet therapy can be an effective second-line intervention, however, the best way to prevent this condition is through counseling of parents or caregivers, and early treatment is essential for optimal therapeutic outcomes.
→The intention of 'elderly' being mentioned here cannot be read from the text. Please remove it from the abstract or provide further details in the text.
RESPONSE: Authors would like to thank the Reviewer for the feedback. We have eliminated the term “elderly” as suggested from our Abstract in lines 25 to 27.
Authors would like to thank again the Reviewer for the valuable feedback and recommendations given. We hope we were able to amend all of the points raised by the Reviewer to improve the reporting of our manuscript.
Reviewer 2 Report
Excellent efforts by the authors, and I am very happy about your quality of the study. However, some minor clarifications are needed from your side to improve it.

It is good
Author Response
Reviewer 2
Excellent efforts by the authors, and I am very happy about your quality of the study. However, some minor clarifications are needed from your side to improve it.
RESPONSE: Authors would like to thank Reviewer for the comments and insightful review. We appreciate the suggestions and we hope we could improve our manuscript according to the review report.
The objective of the study is very vague here. Can you please make it more specific as per your findings.
RESPONSE: Authors would like to thank the Reviewer for the suggestion. We have reformulated our objective adjusting it to our findings.
Are you sure you have restricted your search to only these four databases. Simple tools like google scholar you did not check to verify or any gray literature search you did not perform. Please clarify.
RESPONSE: Authors understand the Reviewer’s concern, and we agree that performing the search in additional databases and addressing gray literature would have enhanced our review scope. We acknowledge this could have made us incur in a search bias, however, we opted to include more than two databases as suggested by systematic review guidelines and chose to include the most relevant ones according to our research topic. Nevertheless, we want to thank the Reviewer for pointing our this matter.
Are all the nine studies included are RCTS. If yes then it should me mentioned some where and if no PEDro can not be used for non RCTs. Needs clarification.
RESPONSE: Authors would like to thank the Reviewer for pointing out this matter. All of the included studies were experimental, however, not all of the studies were RCTs. We have modified our PEDro scale results by eliminating the assessment on non-RCTs, and we have also made amendments in Table 3 to clarify the study type for each included study. Authors appreciate the comment from the Reviewer.
IG and CG also should be abbreviated in the note section
RESPONSE: Authors would like to apologize for the mistake. We have included IG and CG terms in our legend.
Please mention about limitations of your study and future suggestions more clearly at the end of the discussion section.
RESPONSE: Authors would like to thank the Reviewer for the suggestion. We have included a mention about limitations and future research suggestions at the end of our Discussion, in lines 98-107.
Reviewer 3 Report
This is a systematic review and meta-analysis aimed at analyzing conservative treatments employed for the management of Positional Plagiocephaly in infants.
The authors analyzed the available data from nine studies. They indicated that Physiotherapy treatment is considered as the first line of intervention in Plagiocephaly with non-synostotic asymmetries and Manual Therapy is the method that obtains the best results within this intervention. They added that helmet therapy can be an effective second-line intervention in cases of moderate or severe plagiocephaly in elderly patients.
The review deals with an important topic for clinicians, however, there exist some concerns/suggestions that the authors need to address. Please find them below.
1. The authors elaborated on defining plagiocephaly, diagnosis, impact, and incidence in the introduction, while, to contextualize the review, they may need to focus on providing further context and background information about the role of conservative treatments employed for the management of positional plagiocephaly.
2. The study's justification is obscure. The gaps in the body of literature should be pointed out by authors, who should also underline how critical it is to fill such gaps. Furthermore, a stronger justification would also result in a declaration of the potential implications for practice, which is currently missing.
3. Could authors provide a more comprehensive description of the steps used to ensure that specific and reproducible criteria are used to choose the papers that are ultimately included in the review? (i.e., a detailed definition of the inclusion criteria would be interesting)?
4. Why you limited the inclusion criteria of language to “English”? Limiting study inclusion on the basis of language of publication introduces language bias. Both Cochrane and the Campbell Collaboration advise not to restrict the searches by language.
5. When there was a potential source of disagreement between reviewers to decide on study inclusion, the disagreements were resolved by mediation through an adjudicator. How they reached a consensus? The authors should elaborate more on this point in the methodology.
6. Are results from nine studies included in the meta-analysis enough to synthesize the current evidence on the effects of conservative treatments in the management of positional plagiocephaly?
7. Different study designs addressing the same question yield varying results. The risk of presenting uncertain results without knowing for sure the direction and magnitude of the effect hold true for both nonrandomized and randomized controlled trials. In the current study, you integrated multiple study designs. How did you make sure that these study designs were able to address your research question?
8. The discussion reads well. However, it would benefit from further comparison with previous review findings or general literature.
Author Response
This is a systematic review and meta-analysis aimed at analyzing conservative treatments employed for the management of Positional Plagiocephaly in infants.
The authors analyzed the available data from nine studies. They indicated that Physiotherapy treatment is considered as the first line of intervention in Plagiocephaly with non-synostotic asymmetries and Manual Therapy is the method that obtains the best results within this intervention. They added that helmet therapy can be an effective second-line intervention in cases of moderate or severe plagiocephaly in elderly patients.
RESPONSE: Authors would like to thank the Reviewer for the thorough and insightful review. We appreciate the suggestions and comments as they will improve the reporting of our study.
The review deals with an important topic for clinicians, however, there exist some concerns/suggestions that the authors need to address. Please find them below.
- The authors elaborated on defining plagiocephaly, diagnosis, impact, and incidence in the introduction, while, to contextualize the review, they may need to focus on providing further context and background information about the role of conservative treatments employed for the management of positional plagiocephaly.
RESPONSE: Authors would like to thank the Reviewer for the suggestion. We agree that context and background for conservative treatments must be presented to further present the rationale for this study. We have included content in our Introduction section, lines 85-87, to do this.
- The study's justification is obscure. The gaps in the body of literature should be pointed out by authors, who should also underline how critical it is to fill such gaps. Furthermore, a stronger justification would also result in a declaration of the potential implications for practice, which is currently missing.
RESPONSE: Authors thank the Reviewer for the insightful comment. We agree that the justification of the study could be further developed. We have included at the end of our Introduction section, lines 87 to 92, content regarding the rationale of the study, the clinical implication and how important is to better understand the effectiveness of the available conservative treatment options to make better decisions.
- Could authors provide a more comprehensive description of the steps used to ensure that specific and reproducible criteria are used to choose the papers that are ultimately included in the review? (i.e., a detailed definition of the inclusion criteria would be interesting)?
RESPONSE: Authors appreciate the suggestion, and we have furtherly developed our eligibility criteria section trying to clarify the process. The included content can be found in lines 109-117.
- Why you limited the inclusion criteria of language to “English”? Limiting study inclusion on the basis of language of publication introduces language bias. Both Cochrane and the Campbell Collaboration advise not to restrict the searches by language.
RESPONSE: Authors would like to thank the Reviewer for the valuable feedback. We acknowledge that restricting our search to studies published in English introduces a potential language bias, however, we have made efforts to mitigate this limitation by conducting an extensive search for relevant studies within the English-language literature. We have included a statement at the end of the Discussion section mentioning this matter, in lines 100-104. We appreciate the suggestion and we will not restrict searches by language in future studies.
- When there was a potential source of disagreement between reviewers to decide on study inclusion, the disagreements were resolved by mediation through an adjudicator. How they reached a consensus? The authors should elaborate more on this point in the methodology.
RESPONSE: Authors would like to thank the Reviewer for the suggestion. We agree that including a statement about how potential sources of disagreement were handled is important. We have included this information in the corresponding section, in lines 113-114.
- Are results from nine studies included in the meta-analysis enough to synthesize the current evidence on the effects of conservative treatments in the management of positional plagiocephaly?
RESPONSE: Authors understand the Reviewer’s concern about the number of included studies. In the case of positional plagiocephaly and conservative treatments, the current body of evidence is still developing, and the number of eligible studies may be limited. However, we firmly believe that the inclusion of nine studies in our meta-analysis provides valuable insights into the current evidence base. Studies included in our meta-analysis were selected through a rigorous search process that aimed to identify all relevant research available up to our cutoff date. We acknowledge that the inclusion of a larger number of studies would be ideal to further enhance the precision and generalizability of our findings. However, the current state of the research field and the specific focus on conservative treatments for positional plagiocephaly limited the pool of eligible studies.
- Different study designs addressing the same question yield varying results. The risk of presenting uncertain results without knowing for sure the direction and magnitude of the effect hold true for both nonrandomized and randomized controlled trials. In the current study, you integrated multiple study designs. How did you make sure that these study designs were able to address your research question?
RESPONSE: Authors would like to thank the Reviewer for the insightful comment. We acknowledge the potential for differences in methodologies, biases, and sources of variation. However, we believe that by carefully considering the strengths and limitations of each study design and employing appropriate methodological approaches, we were able to account for these differences and derive meaningful insights. We ensured that the selected study designs were able to address our research question by defining a clear research question that encompassed the objectives and outcomes of interest, developed detailed inclusion criteria, conducted a rigorous evaluation of the methodological quality of each included study, and discussed the limitations of our study.
- The discussion reads well. However, it would benefit from further comparison with previous review findings or general literature.
RESPONSE: Authors would like to thank again the Reviewer for the feedback on our discussion section. We appreciate the suggestion to further compare our findings with previous review findings or the general literature. We agree that incorporating additional comparisons would enhance the depth and context of our discussion. Content has been added in lines 97 to 105.
Once again, we sincerely appreciate your insightful comments, which have greatly contributed to improving the quality of our article. We hope that the revisions we have made adequately address your concerns. We look forward to your further evaluation.
Round 2
Reviewer 3 Report
The authors did a good job responding to the earlier concerns and suggestions. The manuscript can be published in its current form.